# You Have Time to Explore Over Here!

## Augmented Reality for Enhanced Situation Awareness in Human-Robot Collaborative Exploration

Terran Mott
terranmott@mines.edu
Colorado School of Mines
Golden, Colorado, USA

Christopher Reardon
christopher.reardon@du.edu
University of Denver
Denver, Colorado, USA

Hao Zhang
hzhang@mines.edu
Colorado School of Mines
Golden, Colorado, USA

Tom Williams
twilliams@mines.edu
Colorado School of Mines
Golden, Colorado, USA

## ABSTRACT

Augmented Reality (AR) is a promising mode of communication for human-robot teaming, due in part to its ability to increase Situation Awareness. In this work, we specifically consider the role of AR in enabling Situation Awareness in time-dominant collaborative human-robot exploration tasks. We present a crowdsourced evaluation of a set of AR visualizations in a simulated version of such a task, and measure the Situation Awareness of experiment participants. Our results raise key questions about the efficacy of Situation Awareness measurement in observational (rather than interactive) evaluation contexts.

## 1 INTRODUCTION AND MOTIVATION

Human-robot teams are valuable in many high-stakes, time-dominant contexts, due to the complimentary abilities of humans and robots [2]. Robots can explore small or dangerous places, while humans can better generalize and adapt. In order for human-robot teams to be effective in time-dominant contexts, such as time-dominant exploration tasks, robots' human teammates need to have high Situation Awareness (SA). That is, they must be aware of their surroundings, be able to use that awareness to create rich mental models of those surroundings, and must be able to use those models to understand and make predictions about the future state of those surroundings. Recently, a variety of researchers have been exploring how Augmented Reality (AR) visualizations may be able to provide such Situation Awareness, by providing enhanced transparency regarding robots' inner workings and intentions (thus increasing humans' SA with respect to those robots), and by directing teammates' attention towards task-critical elements of the environment as perceived by those robots (thus increasing humans' SA with respect to those task-critical environmental features).

In time-dominant contexts, however, it is important for teammates to have high SA not only with respect to their environment and their team, but also with respect to their task context, especially the aspects of their task context that relate to its time-dominant aspects. In time-dominant collaborative exploration contexts, human teammates may have a limited amount of time to explore their environment before needing to reconvene with their teammates. In such contexts, it is critical for humans to maintain awareness of those time constraints, and to understand how their actions may interact with those constraints.

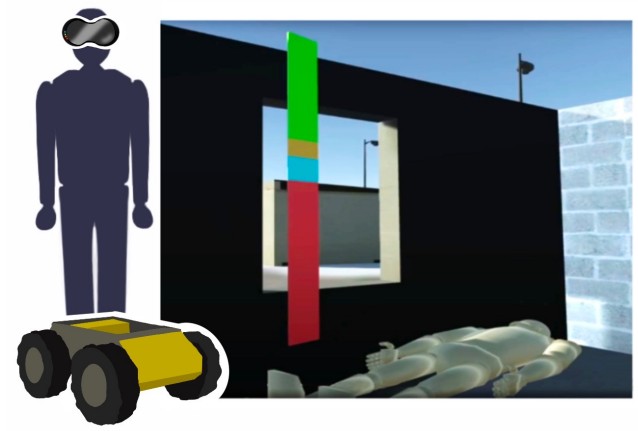

Figure 1: In this paper, we explore Augmented Reality for human-robot communication in a simulated search task.

In this work, we explore how AR may be used to provide exactly this sort of time-oriented SA in time-dominant collaborative exploration tasks. Specifically, we present a set of AR visualizations that depict the interaction between a human's current course of action and/or possible future courses of action with the dynamic time constraints imposed by their task context, and evaluate these visualizations in an observational testbed deployed using the Mechanical Turk crowdsourcing framework. While our results do not provide compelling evidence for the efficacy of our prototyped visualizations, we believe that this is due to the observational nature of our evaluation, as necessitated by COVID-19, and note that this in fact raises key questions about the assessment of SA in observational contexts.

## 2 RELATED WORK

In this section, we will begin by explaining the concept of Situation Awareness (SA), and the role that SA plays within Human-Robot Teaming. We will then describe the ways in which researchers have previously considered SA in the development of Augmented Reality (AR) technologies for Human-Robot Teaming contexts. Finally, we will discuss the different sorts of visualizations (across many types of visual interfaces) that have been used to improve

Situation Awareness across such contexts, and highlight the need to explore new visualizations that specifically aim to increase Situation Awareness with respect to the time-dominant aspects of teammates' situated task contexts.

## 2.1 Situation Awareness

Situation Awareness is defined as a human's "perception of the elements in the environment within a volume of time and space, the comprehension of their meaning, and the projection of their status in the near future" [9]. Models of Situation Awareness were originally developed to understand people working in time-sensitive contexts, including air traffic control [9]. Situation Awareness is divided into three levels of increasing sophistication: perception, comprehension, and projection. *Perception* (Level 1 SA) captures the awareness of basic stimuli in the environment and their properties (e.g., color, shape, and location). *Comprehension* (Level 2 SA) refers to a human's understanding of the meaning of such stimuli. This level includes a human's ability to synthesize their Level 1 observations into a holistic mental model of their environment. *Projection* (Level 3 SA) captures whether a human understands the causal implications of these stimuli (using, e.g., those mental models), in order to form reasonable predictions regarding the near-future state of their environment [9].

As an example, consider a driver who arrives at an intersection. The driver's Level 1 SA (Perception) may be assessed on the basis of whether they perceive that a light has turned yellow. Their Level 2 SA (Comprehension) may be assessed on the basis of whether they comprehend that this is a warning to slow down. Their Level 3 SA (Projection) can be assessed on the basis of whether or not they are aware that if they do not apply the brakes, they are likely to enter the intersection after the light has turned red, risking a collision.

## 2.2 Augmented Reality for Human-Robot Teaming

Augmented Reality (AR) technologies are increasingly being used in Human-Robot Teaming contexts across a variety of task domains [31], including manufacturing, assembly, and surgery [7, 22, 27]. In settings where a human and robot share a physical workspace, AR can help communicate a robot's intended trajectory or action, the hypothetical future location of objects, safety warning messages, and/or general information about the state of the human and robot's shared workspace [1, 3, 6, 7, 22].

As Augmented Reality and 3D Mapping technologies continue to improve, researchers are also increasingly using AR in larger-scale environments that are not limited to shared workspaces, such as collaborative exploration tasks. Collaborative exploration is a broad paradigm of interaction that arises in domains such as search and rescue, disaster response, firefighting, driving, and military operations [12, 18, 25, 33, 38]. These domains are defined not only by their large scale and the focus on systematic team navigation, but also by fundamentally different paradigms of co-location than seen in typical small-scale HRI, with human and robot teammates needing to explicitly and implicitly communicate with each other even when separated by hundreds to thousands of feet.

In order to understand the role that AR (and VR) technologies can play within Human-Robot Teaming contexts, we employ the

Reality-Virtuality Interaction Cube framework [35]. The RVIC argues that AR technologies may enhance interaction in two ways: (1) by enhancing *Flexibility of Control* over humans' robotic teammates, and (2) by enhancing the *Expressivity of View* into those robotic teammates, in order to more effectively build rich mental models.

When viewed through the lens of the RVIC, a number of previous works on AR within large-scale Human-Robot Teaming contexts have focused on enhancing Flexibility of Control. For example, researchers have explored how AR could be used to facilitate new means of teleoperator control over remote robots in fire surveillance and extinguishing tasks [19, 29]. On the other hand, many approaches have also focused on using AR to enhance Expressivity of View, specifically in order to improve Situation Awareness, due to the importance of maintaining SA within the high-stress and dangerous contexts that typically involve collaborative exploration. AR technologies can be particularly effective at enhancing human SA in collaborative exploration contexts by providing increased transparency into the beliefs and perceptions of their robotic teammates, thus increasing human teammates' Level 1 SA by enhancing the inherent perceivability of important stimuli, and increasing human teammates' Level 2 SA by visualizing robots' beliefs and perceptions in a way that directly communicates the *implications* of the beliefs and perceptions that have been selected for visualization.

Indeed, in simulated search and rescue operations, researchers have demonstrated that AR interfaces are uniquely capable for improving the Situation Awareness of robot teleoperators [20]. This has proven particularly useful in the contexts of search and rescue and disaster response, in which real-world human teams often operate with one centralized 'hub' and several decentralized team members [17], providing difficulties for maintaining SA between remote teammates. Moreover, SA is especially critical when robots are involved in such contexts, due to the disparate sensory and navigational capabilities between human and robotic teammates [15]. One recent study found, for example, that many participants teleoperating a robot in a simulated disaster response scenario missed hazardous substances due to insufficient SA [30]. In such cases, AR can help centralized team members to maintain SA with respect to their human and robotic teammates, and can allow centralized teammates to better communicate with their remote teammates in order to enhance those remote teammates' SA.

Moreover, when humans are physically participating in search tasks alongside robotic teammates, AR is a valuable tool to assist human navigation. While human and robot searchers may both be working "in the field" as non-centralized team members, they may in fact be quite distant from their robotic teammates, presenting challenges for information sharing. One study found that AR improved the wayfinding efficiency and task-performance of firefighters navigating a maze-like environment to search for targets [12]. Similarly, in a real-world human-robot cooperative search task, AR allowed humans to successfully navigate to a target identified by a robot [28]. Researchers have speculated that AR may be able to assist disaster response teams by allowing team members (both robots and humans) to share a set of visual 'pins' in the environment to denote dangerous areas and the locations of possible victims [17, 33].

However, in order for SA to be enhanced by AR visualization of robot beliefs and perceptions, visualizations must be carefully

designed in order to selectively draw teammates' attention to task-relevant information that would otherwise have been unattended to (either through passive visualizations or, as in our own work, through active deictic communication strategies [13, 34, 36]), without distracting teammates and thus causing them to ignore other critical task-relevant information. Researchers employing AR visualizations for human-robot teaming contexts have proposed a number of methods for ensuring that visualizations increase SA without cognitively and perceptually overloading robots' human teammates [8, 14]. For example, researchers have suggested that members of human-robot search and rescue teams be shown only visualizations of information directly relevant to their personal goals and circumstances [4]. Similarly, others have shown how information displayed in maps used by UAV operators can be carefully filtered to facilitate better Situation Awareness by retaining and emphasizing only the most salient information [5, 37].

Most of the approaches above, however, have focused on either enabling new methods of control over humans' robotic teammates, or on visualizing information perceived by those robotic teammates. However, a variety of information may be helpful to visualize in time-dominant contexts that leverages but is not completely informed by the robot's own perceptions. In time-dominant exploration contexts, for example, one of the key things that human explorers must be cognizant of is the time-dominant aspects of their task (e.g., how much time remains until they need to meet up with their teammates). Critically, these task-oriented concerns may interact with a human teammate's personal situation and with the information gathered by robotic teammates in unique ways.

In recent work [26], we proposed and prototyped four types of visualizations that communicate information to human teammates about (1) where they need to be when the time allotted to their task expires; (2) how much of that allotted time remains, and the expected proportion of that remaining time required to reach that target location; (3) how much of that allotted time is expected to remain if they continue exploration in a given direction, on the basis of information collected by robot teammates; and (4) how much of that allotted time is expected to remain if they engage with a particular task-relevant object located by their robotic teammates. These visualizations are thus designed with the aim of enhancing the time-oriented SA of robots' human teammates, by combining information stemming from the general task context (i.e., amount of remaining time and location of meetup point), information specific to the human's situation (i.e., relative to how far they currently are from that meetup point), and information available due to the observations of their robotic teammates (i.e., information regarding task-relevant objects found by robotic teammates).

In the next section, we will present these four visualizations. In the following section, we will then present the results of a human-subject experiment designed to assess the ability of these visualizations to enhance human teammates' SA as intended.

# 3 VISUALIZATION DESIGN

Before describing the four visualizations we designed to enhance time-oriented SA, we must first describe the context in which these visualizations were intended to be used.

## 3.1 Interaction Context

Our visualizations are designed to be used in time-dominant exploration tasks. To this end, we created a simulated search task in which a human navigates a simulated environment in search of different task-relevant objects, such as searchable filing cabinets and treatable victims. The human's goal in this context is to evaluate as many objects as possible within the constraints of the task.

This context is time-dominant due to the confluence of two factors. First, the human operating in this environment has a limited amount of time to explore the environment, and must reach the meet-up point by a particular time. Second, successfully interacting with any of the targets found in the environment requires investment of a set amount of time. In addition, the time-dominance of the task involves an element of loss that exceeds diminishing returns. Reaching the meet-up point late may have adverse consequences, especially in domains like emergency and disaster response.

The human exploring this environment is accompanied by a robot teammate, whose aim is to improve the human teammate's SA by scouting the environment and informing its human teammate of the locations of targets and/or the amount of time required to interact with them. The robot achieves this through visualizations displayed in the human teammate's Augmented Reality Head-Mounted Display (AR-HMD). In the present work, these visualizations were simulated as overlays over a user's perspective in a Virtual Reality simulation designed in the Unity-based DCIST simulator developed by the U.S. Army Research Laboratory through the *Distributed and Collaborative Intelligent Systems and Technology* (DCIST) Collaborative Research Alliance [16].

## 3.2 Interaction Design

In this section, we describe the different visualizations rendered by the robot in its human teammate's AR-HMD (the design process used to prototype and implement these visualizations is described in [26]).

### Phase Line Visualization

The first visualization we developed was a *phase line*, which marks the area that the human must return to by the end of their allocated time. The phase line appears as a ring of concentric cyan circles ascending into the sky. Its design was adapted from traditional phase line symbols used by the US military as map annotations to coordinate team movement [26]. While phase lines are typically depicted as two-dimensional shapes, our phase line was also given a verticality in order to be viewed from far away [26]. This visualization is designed to increase SA by helping users to understand how far away their team's meet-up point is, and a frequently visible reminder of that point, and thus of the time constraints connected to that point.

### Timebar Visualizations

The other three SA-enhancing visualizations used in our experiment are all versions of a *timebar*, and are specifically designed to enhance temporal rather than spatial SA. As shown in Fig. 3, timebars are rectangular visualizations that depict aspects of a task's time sensitivity using a consistent color scheme.

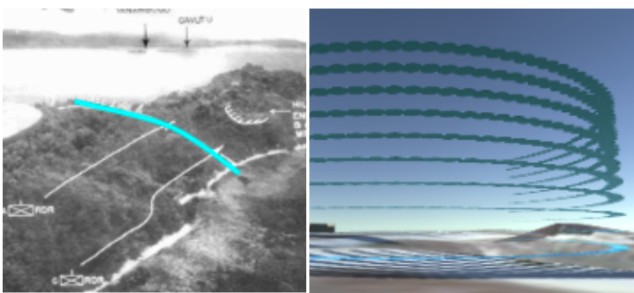

Figure 2: A traditional military phase line drawing on a map (left). The 3D phase line visualization (right).

**Overall task time visualizations:** The overall task time visualization used is an *overall task timebar* 3 that appears on the left-hand side of the human's vision and showed information about the amount of time remaining in the task. This visualization was intended to enhance users' general Situation Awareness regarding the time constraints of the task.

**Summary time-cost visualizations:** The summary time-cost visualization used is a *summary time-cost timebar* 3, which appears at the entrance to buildings the robot has explored and shows the time demands needed *if* one were to interact with all targets in that building. This visualization is intended to enhance users' Situation Awareness surrounding the interaction between those time constraints and the potential courses of action available in a particular building.

**Target time-cost visualizations:** The target-specific time-cost visualization used is a *target summary timebar* 3, which appears above targets the robot has found and shows the specific time requirements of interacting with those targets. This visualization is intended to enhance users' Situation Awareness surrounding the precise causal outcomes of specific courses of action.

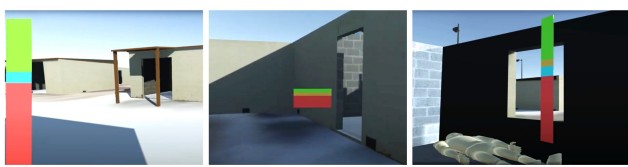

Figure 3: (left to right) The task timebar, summary time-cost timebar, and target time-cost timebar as they appear in the task.

The colors used in all timebar visualizations sent by the robot teammate have consistent meanings. In all cases, the red portion represents the time elapsed in the task. The green represents remaining time available to evaluate targets. As time passes, the red section grows and the green secction shrinks. In the building and target summary time-cost timebars, the yellow section represents the portion of remaining time that would be required for the human to evaluate the individual target (in the case of target time-cost

timebars) or all the targets in the building (in the case of summary time-cost timebars). The blue represents exit time, i.e., the portion of remaining time required to navigate to the phase line (it changes as the human gets closer to or farther from the phase line). Summary time-cost timebars do not visualize exit time because it is unclear where a human will choose to exit a building if they choose to search it.



Figure 4: The overall task timebar before (middle) and after (right) interacting with a target.

## 4 EXPERIMENTAL METHODS

In order to assess the effectiveness of these visualizations at enhancing SA, we conducted a human-subject experiment conducted using the aforementioned DCIST simulator. Specifically, this experiment was designed to test the following four hypotheses.

### 4.1 Hypotheses

**H1:** The timebar visualizations that convey the time-needs and potential time-costs of various task characteristics will promote better Situation Awareness by providing information about the aspects of the tasks that are most critical to its time-dominant nature.

**H2a:** The overall task timebar will result in higher overall Situation Awareness because it provides a constant visualization of high-level time-dominant task characteristics, like exit time.

**H2b:** The summary time-cost visualization will result in higher level 3 Situation Awareness because visualizing the cost of potential actions will help participants better predict the actions of the hypothetical human teammate in the video.

**H2c:** The target summary time-cost visualization will result in higher level 2 Situation Awareness because visualizing the implications of the decision to evaluate a specific target will help participants understand why the human teammate in the video chooses to evaluate a target.

## 4.2 Experimental Design

The experiment conducted to evaluate these hypotheses followed a 2x2x2 between-subject design. Participants observed a task in which visualizations designed to facilitate Situation Awareness through three strategies were either shown or not-shown.

Specifically, the *Phase Line Visualization* was used across all experimental conditions to consistently emphasize the use of AR in the experimental task, and the *Overall task time visualizations*, *Summary time-cost visualizations*, and *Target time-cost visualizations* were each either shown or not shown depending on experimental condition.

## 4.3 Procedure

Due to the safety restrictions imposed by COVID-19 [10], we conducted our experiment online, with participants watching videos filmed within our experimental environment rather than navigating the environment themselves. After giving informed consent, participants read instructions describing the meaning of the visualizations they would see. Then, they watched a short video recorded in our simulation environment. This video ended before the completion of the task. After completion of this task, the participants completed a battery of survey elements in order to assess the effects of experimental condition on Situation Awareness.

## 4.4 Measures

To measure Situation Awareness, we used two sets of survey questions based on existing explicit and subjective measures.

Our first survey was comprised of explicit methods to directly measure Situation Awareness at each level (perception, comprehension, projection), modified from the QASAGAT (Quantitative Analog Situation Awareness Global Assessment Technique) which is often used to measure SA via the *freeze* technique of interrupting a task and immediately asking for answers [11]. We approximated this by ending the video before time ran out on the task, and then immediately administering this survey. For each Situation Awareness level, participants answered two questions:

**Level 1 SA: Perception**
(1) Which of these locations was the human's location at the end of the video? (picture provided)
(2) How many targets did the human find and evaluate in the video?

**Level 2 SA: Comprehension**
(1) Which of these paths did the human take to find their first target? (picture provided)
(2) When the video ended, where was the phase line relative to where the human was facing? (multiple choice)

**Level 3 SA: Projection**
(1) At the end of the video, do you think the human could have made it to the phase line before time ran out?
(2) How many seconds do you think the human would need to return to the phase line at the end of the video?

Our second survey was adapted from the SPASA rating scale (Short Post-Assessment of Situation Awareness) of Likert items on a scale of (strongly disagree, disagree, agree, strongly agree) [11]. The SPASA questions we asked were whether or not participants agreed with each of the following statements:

| $BF_{Incl}$ | O | S | T | OxS | SxT | OxT | OxSxT |
|---|---|---|---|---|---|---|---|
| *Overall SA* | .180 | .282 | .671 | .216 | .218 | .203 | .304 |
| *Level 1 SA* | .257 | .162 | .629 | .599 | .262 | .200 | .388 |
| *Level 2 SA* | .217 | .240 | .153 | .201 | .203 | .224 | .310 |
| *Level 3 SA* | .190 | .528 | .365 | .236 | .242 | .360 | .237 |
| *Subjective* | .152 | .147 | .208 | .208 | .631 | .218 | .286 |

**Table 1: Effects by visualization type(s). O: overall task timebar. S: time-cost summary timebar. T: target time-cost timebar.**

(1) It was easy to keep track of the time aspects of the task.
(2) It was easy to predict what the human would choose to do next.
(3) The information in the visualizations was provided at a rate that I could easily perceive.
(4) I had a good overall understanding of the human's situation during the task.

We computed an overall Situation Awareness score based on the number of QASAGAT questions a participant answered correctly, as well as scores based on their correctness at each level. We also numerically coded and averaged participants' SPASA responses into an overall subjective assessment score.

## 4.5 Participants

Data was collected from 218 participants using Amazon's Mechanical Turk platform. 136 participants identified as male, 81 identified as female, and 1 declined to report their gender identity. The average age of participants was 37 with a standard deviation of 10.9.

## 4.6 Analysis

To analyze the effects of AR visualizations on Situation Awareness, we performed a series of Bayesian Analyses of Variance (ANOVAs), with Level 1 SA, Level 2 SA, Level 3 SA, Overall SA, and participants' averaged subjective scores as dependent variables, and the use of the overall task visualization, summary time-cost visualizations, and target time-cost visualizations as independent variables [24]. These ANOVAs were followed by calculation of Bayes Inclusion Factors across matched models [21, 23] to determine the impact of each visualization, with post-hoc pairwise t-test analyses performed for any results with no more than 3:1 ($BF_{Incl} = 0.333$) against an effect. These analyses were performed using the JASP statistical analysis software [32].

## 5 RESULTS

As shown in Tab. 1, our results were uniformly negative or inconclusive. Here, inconclusive results suggest that more data must be collected before our hypotheses can be confirmed or refuted.

*Overall SA* — Our analysis of the effect of visualizations on overall Situation Awareness provided inconclusive evidence regarding an effect of target time-cost visualizations on overall SA ($BF_{Incl} = 0.671$), providing sufficiently weak evidence against an effect that more data would need to be collected before confirming or

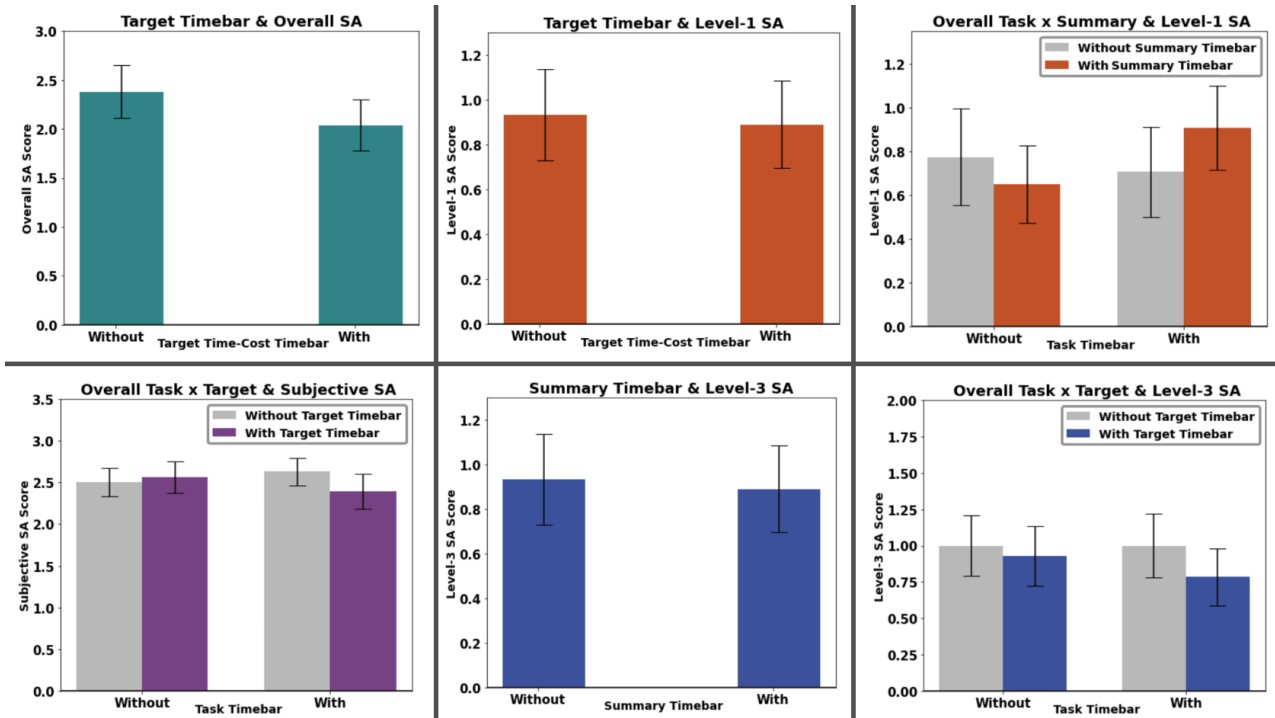

Figure 5: Visualization of Experimental Results

ruling out an effect. For all other main and interaction effects, moderate evidence was found against an effect ($0.1 < BF_{incl} < .333$) in all cases.

*Level 1 SA* — Our analysis of the effect of visualizations on Level 1 SA provided inconclusive evidence regarding effects of target time-cost visualizations on Level 1 SA ($BF_{Incl} = 0.629$), and regarding an interaction effect between overall task and summary time-cost visualizations on Level 1 SA ($BF_{Incl} = .599$). Post-hoc pairwise t-test analyses of this potential interaction effect specifically showed inconclusive evidence regarding a difference between simultaneous use of overall task and summary time-cost visualizations and use of summary time-cost visualizations ($BF 1.129$) or overall task ($BF = 0.517$) visualizations alone, and moderate evidence against all other effects.

*Level 2 SA* — Our analysis of the effect of visualizations on Level 2 SA provided moderate evidence against an effect of any visualization type on Level 2 SA.

*Level 3 SA* — Our analysis of the effect of visualizations on Level 3 SA provided inconclusive evidence regarding effects of summary time-cost visualizations ($BF_{Incl} = 0.528$) and target time-cost visualizations ($BF_{Incl} = 0.365$) on Level 3 SA, and inconclusive evidence regarding an interaction effect regarding the use of summary time-cost visualizations and target time-cost visualizations on Level 3 SA ($BF_{Incl} = 0.360$).

*Subjective Self-Assessment of SA* — As shown in Tab. 1, our results were uniformly negative or inconclusive. We found inconclusive evidence regarding the effect of the interaction between summary time-cost and target time-cost visualizations on participants' subjective score ($BF_{Incl} = 0.631$). This is sufficiently weak evidence against an effect and would require the collection of more data.

## 6 DISCUSSION

The results of this study were uniformly negative or inconclusive, meaning that the AR visualizations used in our work did not appear to provide any benefit to SA within our experimental paradigm. As such, our evidence fails to support hypothesis **H1**, that the visualizations would promote Situation Awareness. In fact, our inconclusive result with respect to overall SA benefits were found in the case of the target time-cost timebar, for which if evidence of an effect were revealed by further data, the revealed effect would most likely be a *negative* impact of the target time-cost timebar on overall SA.

Similarly, our evidence failed to support hypotheses **H2a**, **H2b**, and **H2c**, which stipulated that particular visualizations would improve specific levels of Situation Awareness. In fact, the only inconclusive evidence which would be expected to reveal evidence *in favor* of the use of any of the presented visualizations, if an effect were found, was in the case of the overall task timebar and the summary time-cost timebar, which *may* have had a positive impact, but *only* on Level 1 SA, and *only* when used together.

These results likely have much to do with the fact that participants in this study merely watched a video of a simulated task; they were not responsible for making any decisions during the experiment. Fundamentally, this lowers the stakes of the activity. Participants only had to observe (and possibly speculate about)

the task environment. In this sense, Situation Awareness was less necessary (perhaps even *un*necessary) due to the passive nature of the task. Situation Awareness is not automatically present in a task; it is acquired and maintained [9]. Participants did not *need* Situation Awareness to successfully watch a video in the way they would need it to complete an active task. So, it is probable that many participants did not build or maintain Situation Awareness during the task.

These results are perhaps unsurprising when we consider that Endsley's three-level framework for Situation Awareness was not designed or intended to evaluate passive observation tasks. Even the original settings in which Situation Awareness was explored (like air traffic control) involve active human decision making [9]. Indeed, there are fundamental differences between doing a task and watching someone else do a task. In light of this, the results of this study should not be used to speculate about the value of the AR visualizations for those who may actively complete our simulation. Due to COVID-19, it was not feasible to conduct this experiment in-person, where participants could have done the exploration task themselves. When it is safe to do so, it would be worthwhile to conduct a similar experiment in which the same visualizations (and their combinations) are presented to participants in the context of a live, interactive, human-robot teaming scenario (even if conducted within a simulator such as that used in this work). In such a follow-up experiment, participants would need to build and rely on Situation Awareness in a much more substantial way.

## 7 CONCLUSION

In this paper we presented a set of AR visualizations designed to increase time-relevant Situation Awareness in human-robot collaborative exploration tasks. Moreover, we present the results of a user study intended to evaluate the effectiveness of these visualizations at building SA. While our results were negative or inconclusive, we intend in future work to replicate this experiment in a live interaction scenario, which we have argued would be substantially better suitable due to our focus on Situation Awareness.

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
