# OpenReview forum: "You Have Time to Explore Over Here!: Augmented Reality for Enhanced Situation Awareness in Human-Robot Collaborative Exploration"
_humanrobotinteraction.org/HRI/2021I/Workshop/VAM-HRI — VAM-HRI 2021 Oral_

### Official Review · AnonReviewer2 · 2021-03-04
**Enhancing situational awareness using augmented reality for time-sensitive robot tasks**

**Rating:** 8
**Confidence:** 5

**Review:**

This paper investigates the effectiveness of augmented reality (AR) for improving human situational awareness in time-sensitive collaborative robot domains involving explorative tasks. This work uses AR visualizations to depict a human’s course of actions and possible future courses of actions and the time constraints associated with these decisions based on their task context. This approach is novel because previous works have not investigated how visual information in AR can be used to communicate situational awareness specifically in time-dominant exploration contexts where timing is key. This paper conducts a study on Amazon Mechanical Turk with 218 participants using the DCIST simulator, with a task where users watched a video of someone scouting around an environment searching for different task-relevant objectives, with the goal of evaluating as many objects as possible within the time constraint. Users were shown four different visualizations, one regarding where the human needed to go by the end of the task and three others depicting different time-sensitive information. This paper made four different hypotheses regarding how they predicted these visual indicators would positively impact situational awareness. The results were either inconclusive or negatively supported all the hypotheses made.

As the authors mention, the fact that this experiment was done in an observational capacity rather than letting users explore the space themselves may well explain why the results are negative and inconclusive. This indicates that when possible, experiments should be redone with real participants in order to correctly evaluate the effects of these visualization methods. Nevertheless, this paper presents an interesting problem that has not been well explored, and conducts good experimental design and comprehensive analysis on the results. The related work is well-stated, and the author particularly enjoyed the description of situational awareness and the motivating importance of it in these time-sensitive tasks. Overall, I recommend this paper be accepted.

Some comments and questions:
- Was only color used to indicate time, or was the actual amount of time also visualized? How would visualizing the exact amount of time left impact SA?
- How does the shape of the timing visualizer affect SA? What if instead of a bar, a clock shape was used instead? would this be more intuitive?
- Could sound be used in an effective way to convey time-sensitive information? For example, a ticking sound every second could provide users with time-sensitive information when they are not directly looking at the visualizations.

---

### Official Review · AnonReviewer1 · 2021-03-08
**Incomplete but interesting work, room for defining vocabulary for task level interactions**

**Rating:** 7
**Confidence:** 4

**Review:**

Interesting work, though the results are largely inconclusive, and the experimental setup is quite limited given the current circumstances. A couple of comments:

1. In terms of modeling situational-awareness, especially as it relates to tasks, one problem that arises is the model of the teammate itself and how that differs from each other (e.g. their situational awareness will be different and that will affect their expected response which in turn affects teaming). Especially when navigating such issues, as well communicating about task information, it is essential to have a well-defined vocabulary so that the implications of a symbol on the task can be precisely computed and hence communicated e.g. in [1]. This is especially critical since symbols are lossy in terms of what aspects of a task it can describe, once the authors move beyond task completion and deadlines.

2. On the topic of vocabulary and search & reconnaissance tasks -- I would recommend the authors to investigate established standards in such tasks like in [2] [3] which offer a very rich vocabulary of symbols to communicate indirectly with teammates. Instead of just time and task remaining, you could map these symbols for visualization to what task-level information they can provide. Interestingly, for mixed-reality, one "advantage" is that events need not have happened (or the robot need not to have navigated to a spot) to put up symbols (previously on the walls). So this allows for 1) a more flexible vocabulary as well, in addition to 2) utilizing that advantage toward affecting better teaming.

[1] Chakraborti, et al. Projection-aware task planning and execution for human-in-the-loop operation of robots in a mixed-reality workspace. IROS 2018.
[2] https://www.fema.gov/pdf/emergency/usr/usr_23_20080205_rog.pdf
[3] http://www.fire-vols.org/Documents/building_marking_system.pdf

Of course, doing such things in mixed-reality assumes access to infrastructure (such as network availability) that may not be there at a place where the search and rescue operation is taking place.

As an aside, organizationally, I felt it would work better if the notions of SA, then the metrics to measure it for the task at hand, and then the hypothesis, came together and in that order, rather than being spread across the paper.

---

### Decision · Program_Chairs · 2021-03-06

Accept (Oral)